# Understanding Drivers of Glacier Length Variability Over the Last Millennium

Alan Huston[1], Nicholas Siler[1], Gerard H. Roe[2], Erin Pettit[1], and Nathan J. Steiger[3,4]

[1]College of Earth, Ocean, and Atmospheric Sciences, Oregon State University, Corvallis, OR, USA
[2]Department of Earth and Space Sciences, University of Washington, Seattle, WA, USA
[3]Institute of Earth Sciences, Hebrew University, Jerusalem, Israel
[4]Lamont-Doherty Earth Observatory, Columbia University, Palisades, NY, USA

**Correspondence:** Nicholas Siler (nick.siler@oregonstate.edu)

**Abstract.** Changes in glacier length reflect the integrated response to local fluctuations in temperature and precipitation resulting from both external forcing (e.g., volcanic eruptions or anthropogenic $CO_2$) and internal climate variability. In order to interpret the climate history reflected in the glacier moraine record, therefore, the influence of both sources of climate variability must be considered. Here we study the last millennium of glacier-length variability across the globe using a simple dynamic glacier model, which we force with temperature and precipitation time series from a 13-member ensemble of simulations from a global climate model. The ensemble allows us to quantify the contributions to glacier-length variability from external forcing (given by the ensemble mean) and internal variability (given by the ensemble spread). Within this framework, we find that internal variability is the predominant source of length fluctuations for glaciers with a shorter response time (less than a few decades). However, for glaciers with longer response timescales (more than a few decades) external forcing has a greater influence than internal variability. We further find that external forcing also dominates when the response of glaciers from widely separated regions is averaged. Single-forcing simulations indicate that, for this climate model, most of the forced response over the last millennium, pre-anthropogenic warming, has been driven by global-scale temperature change associated with volcanic aerosols.

## 1 Introduction

The length of a glacier reflects both its topographic and climate setting, and arises from a balance between accumulation (mass gain) and ablation (mass loss), mediated by the catchment geometry and the dynamics of ice flow. Past fluctuations in glacier length often leave moraines on the landscape. If the age of these moraines can be determined, they become valuable proxies of past climate change (e.g., Solomina et al., 2015). The decadal-and-longer response time of glacier length means that glaciers act as low-pass filters of climate, and thereby have the potential to reveal climate trends that would otherwise go undetected (Balco, 2009).

Changes in climate can arise from three factors. First, natural climate forcings, such as solar luminosity, variations in the Earth's orbit, and volcanic eruptions that alter the fluxes of energy entering or leaving the climate system; second, anthropogenic forcing, such as emissions of $CO_2$ and industrial aerosols, that also affect Earth's energy budget; and third, the internal climate

variability that would occur in the atmosphere-ocean-cryosphere system even under constant external conditions. Internal
variability can be thought of as the year-to-year fluctuations that happen due to the chaotic nature of atmospheric and oceanic
dynamics. It can exhibit some degree of interannual persistence due to the longer-timescale components of the system (see
Burke and Roe, 2014, for a discussion in the context of glaciers).

Our primary focus in this study is on the relative importance of forced and unforced climate variability for glacier fluctuations
in the pre-industrial era. We focus on the pre-industrial era because of its importance in interpreting the Holocene glacier record.
Secondly, for the industrial era (since about 1880 or so), the central estimate of the warming from anthropogenic forcing is
one-hundred percent of the observed warming, both globally and regionally (Haustein et al., 2019). This implies that both
the observed industrial-era mass loss (Roe et al., 2020) and retreat (Roe et al., 2017) of Alpine glaciers is overwhelmingly
anthropogenic in origin.

Until recently, variability in glacier length over the pre-industrial Holocene was mostly attributed to natural forcing of
35 temperature. For example, Crowley (2000) estimated that 40-to-65 % of pre-industrial decadal-scale temperature variations
were a result of changes in solar irradiance and volcanism. Similarly, Miller et al. (2012) have argued that the Little Ice Age
period between approximately 1300 and 1850 was caused by a sequence of volcanic eruptions, with the cooling effects of
explosive volcanism made more persistent by sea-ice/ocean feedbacks operating long after the volcanic aerosols were removed
from the atmosphere.

However, recent studies have shown that large variability in glacier length can also occur in an unforced, statistically constant
climate due to internal climate variability (e.g., Oerlemans, 2000). By studying glaciers around Mt. Baker in Washington State,
for example, Roe and O'Neal (2009) found that internal variability alone can produce kilometer-scale excursions in glacier
length on multi-decadal and centennial timescales. This result highlights the importance of considering internal variability in
addition to forced climate change as a potential cause of past variations in glacier length. Until now, however, no systematic
assessment of the relative importance of forced versus internal variability has been conducted.

Both the local magnitude and the spatial coherence of glacier-length variability have been studied for specific regions using
existing observations. For example, there is a notable inter-hemispheric disparity in the timing of the maximum ice extent
during the Holocene for glaciers in New Zealand versus the Alps (Schaefer et al., 2009). Additionally, mid-to-late Holocene
glacier fluctuations were neither in phase nor in strict anti-phase between the hemispheres, suggesting that regional climate
variability has played an important role (Schaefer et al., 2009). Indeed, the very idea of a global-scale Little Ice Age or Medieval
Climate Anomaly has been called into question (Neukom et al., 2019).

We evaluate the relative importance of forced and unforced climate variability on driving glacier-length fluctuations using
two primary research tools. The first is an archive of ensemble simulations of a global climate model, from which we make
estimates of the forced and unforced components of the climate variability, as well as the impact of each type of climate
forcing. The second is a simple dynamical glacier model whose parameters can be adjusted to different climate settings and
glacier geometries. We measure the relative importance of forced and unforced variability using the signal-to-noise ratio (SNR,
defined in the next section), which has been used before in the context of glaciers in Anderson et al. (2014) and Roe et al. (2017).

We begin by applying the glacier model to three widely separated locations with different climatic settings. The analyses highlight the importance of glacier response time: short response time (i.e., less than a few decades) glaciers have uncorrelated length fluctuations driven predominantly by regional, internal climate variability (a low SNR); whereas longer response-time glaciers respond coherently to external changes in climate forcing, predominantly associated with volcanic forcing (a high SNR). Finally, we extend our analyses to a global network of 76 well-observed glaciers to assess the coherence of glacier response among, and within, individual glacierized regions.

## 2   Quantifying forced and internal climate variability over the last millennium

We analyze climate variability in the Community Earth System Model (CESM) Last Millennium Ensemble (LME). The LME comprises 13 simulations that span 850-2005 CE (Otto-Bliesner et al., 2016). Each simulation includes the same radiative forcing contributions from volcanic aerosols, solar irradiance, orbital changes, greenhouse gases, and ozone aerosols, based on forcing reconstructions from the fifth-generation Coupled Model Intercomparison Project (CMIP5; Schmidt et al., 2011). The ensemble members all have exactly the same physics, and differ only in their initial conditions. Chaotic internal dynamics and the different initial conditions cause a spread among the ensemble members. Hence, the mean of the ensemble is an estimate the forced climate response, while the spread among ensemble members represents internal climate variability (e.g., Deser et al., 2012).

For the first part of our analysis, we focus on three geographically diverse glaciers, with distinct climatic settings: i) the Silvretta glacier in the Alps, ii) the South Cascade glacier in the American Pacific Northwest, and iii) the Martial Este glacier in the Patagonian Andes. Each of these has good observations of its mass balance over the last few decades (Medwedeff and Roe, 2017, and Table A1).

We identify the grid points in the CESM model that are closest to each glacier. From the archived model output at each of these grid points, we compute annual time series of average winter precipitation ($P$) and summer temperature ($T$). For grid points in the Northern Hemisphere, $P$ is taken from October-March and $T$ is taken from April-September. The seasons are flipped for grid points in the Southern Hemisphere. We then estimate mass-balance variability using the following linear approximations,

$$b'_w = \alpha P' \tag{1}$$

and

$$b'_s = -\lambda T', \tag{2}$$

where $b_w$ and $b_s$ represent winter and summer mass balance, $\alpha$ and $\lambda$ are positive constants, and primes indicate anomalies relative to the millennial average (1000-1999). Note that the negative sign in Eq. 2 reflects an anticorrelation between mass balance and temperature in the summer.

Like most GCMs, the grid resolution of the LME is too coarse to capture the full influence of orography on $P$ and $T$ at most glacier locations. To compensate for this, we choose values of $\alpha$ and $\lambda$ for each glacier that produce variances in $b'_w$ and $b'_s$ that

match the observed values reported by Medwedeff and Roe (2017). This is achieved by setting

$$\alpha = \frac{\sigma_{b,w}}{\sigma_P} \tag{3}$$

and

$$\lambda = \frac{\sigma_{b,s}}{\sigma_T}, \tag{4}$$

where the numerators represent the observed standard deviations in winter and summer mass balance, and the denominators
represent the standard deviations in winter precipitation and summer temperature.

$$b' = b'_w + b'_s. \tag{5}$$

Figure 1 shows the mass-balance time series created in this way. Their average is shown in the lowest panels. The vertical grey line at the year 1880 marks the approximate transition from the pre-industrial era to the modern era, after which the time axis on each panel is dilated by a factor of 4 to enhance visibility. Figure 1 also shows $b'_s$ calculated from observed temperature
anomalies (NASA GISS dataset; Lenssen et al., 2019).

The importance of the external forcing relative to the internal variability can be measured by a signal-to-noise ratio (SNR), which we define as

$$\text{SNR} = \frac{\text{Var}[\overline{b'}(t)]}{\text{Var}[b'(t)] - \text{Var}[\overline{b'}(t)]}, \tag{6}$$

where $\text{Var}[\overline{b'}(t)]$ is the variance of the ensemble mean time series (i.e., the "signal" of the forced response), and $\text{Var}[b'(t)]$ is
105 the total variance across all ensemble members (i.e., as if all 13 time series were concatenated into a single time series). The difference between the two quantities in the denominator represents the variance due to internal climate variability (i.e., the "noise"). In general, SNR values less than one indicate that most of the variance in a time series can be attributed to internal variability, while SNR values greater than one indicate that most variance is driven by external forcing. For a 13-member ensemble, SNR values greater than 0.125 in the modern era or 0.098 in the pre-industrial era indicate that a forced signal can
be detected and is statistically significant at the 99% confidence level (see Appendix). SNR values are listed in Fig. 1 for each time series and time period, with bold font indicating statistical significance.

In the winter (Fig. 1a), there is little evidence of a forced signal in $b'_w$ at any glacier during either time period, as indicated by SNR values that are uniformly low and statistically insignificant. This implies that interannual variability in winter mass balance (and thus winter precipitation) at these glaciers is dominated by internal climate noise. This finding is expected based
on past studies of the contribution from winter precipitation trends in glacier change (e.g., Medwedeff and Roe, 2017).

In contrast, $b'_s$ shows clear evidence of a forced signal (Fig. 1b). During the pre-industrial period, all locations exhibit several abrupt spikes associated with volcanic eruptions, with the most notable event being the Samalas eruption in 1257 (e.g., Guillet et al., 2017). The amplitude of these spikes is strongest in Europe and North America, where they occur against the backdrop of a longer-term increase in $b'_s$ associated with a cooling trend that persists through the end of the 19th century. Although the

forced signal is statistically significant at all locations during the pre-industrial period, SNR values are less than 0.2, implying that internal noise still accounts for the majority of local year-to-year temperature variability.

Around 1900, $b'_s$ begins to decrease at all three glaciers in response to global warming (Fig. 1b). SNR values are greater in this period than in the pre-industrial period, reflecting the unprecedented strength and persistence of anthropogenic forcing. Comparing the simulated trends in $b'_s$ over the 20th century with those derived from observed temperatures (Fig. 1b; colored vs. black lines), we find good agreement at South Cascade and Martial Este but not at Silvretta, where the decrease in $b'_s$ in the LME ensemble mean is too low by 0.07 m/decade, reflecting a local warming trend that is too low by 0.063 K/decade (Eq. 2, with $\lambda = 1.11$ m K$^{-1}$). One possible interpretation of this difference is that part of the observed warming trend at Silvretta was the result of internal variability. This conclusion is supported by the fact that the observed trend at Silvretta lies within the ensemble range of simulated trends. On the other hand, 20th-century warming trends in the LME are weaker than observed trends over most of the globe (Fig. A1), suggesting that model bias likely also plays a role. We consider possible causes of this bias in Section 3.4, but note that it does not affect the bulk of our analysis.

Finally, variability in $b'$ (Fig. 1c) shows the influence of both $b'_w$ and $b'_s$, but in different proportions for each glacier. For example, at South Cascade glacier (red), the variance in $b'_w$ is a factor of 2 greater than the variance in $b'_s$, and thus exerts a much stronger influence on $b'$. Because $b'_w$ is essentially noise, the SNR of $b'$ at South Cascade glacier is statistically insignificant in the modern era, even though $b'_s$ contains a significant forced signal. This masking of forced variance in $b'$ by large internal variance in $b'_w$ despite the presence of forced variance in $b'_s$ is typical of maritime glaciers and detailed, for example, in Young et al. (2020). At Silvretta and Martial Este, by contrast, the variance comes mostly from $b'_s$. Thus, while $b'_w$ remains a source of noise, its amplitude is small enough that the forced signal from $b'_s$ can still be detected in $b'$.

A last point to emphasize from Fig. 1 relates to the averaged time series among the three glaciers (green lines). These exhibit higher SNRs in $b'_s$ and $b'$ than in any individual glacier: spatial averaging suppresses noise and brings out a common forced signal. This echoes results from studies of modern-day warming, which have found that the anthropogenic signal is clearest at global scales, even when it is obscured by internal variability at local and regional scales (e.g., Deser et al., 2012). We elaborate on the reasons for this result in Section 3.2 below.

## 3 Glacier simulations

The implications of this climate variability for fluctuations in glacier length are evaluated using the three-stage linear model of Roe and Baker (2014), which has been shown to better capture the high-frequency response of glacier dynamics than earlier low-order models (Harrison and Post, 2003; Oerlemans, 2000, 2005). The three stages, which can be diagnosed from ice dynamics in numerical models, are: (1) changes in interior thickness, which drive (2) changes in terminus ice flux, which in turn drive (3) changes in glacier length. Collectively, the three stages can be represented as a linear, third-order differential equation,

$$\left(\frac{d}{dt} + \frac{1}{\epsilon\tau}\right)^3 L' = \frac{1}{\epsilon(\epsilon\tau)^2}\beta b', \tag{7}$$

where $L'$ is the length anomaly as a function of time, $t$; $b'$ is the annual mass balance anomaly derived from LME output; $\epsilon$ is the ratio of the variances in the three-stage and one-stage models, which is set to $1/\sqrt{3}$; $\tau$ is the glacier response timescale, and $\beta$ is a non-dimensional shape parameter that only affects the amplitude of $L'$, and not its temporal characteristics. The glacier response time is given by

$$\tau = -H/b_t, \tag{8}$$

where $H$ is a characteristic ice thickness in the terminus zone and $b_t$ is the net (negative) mass balance in the terminus zone (Jóhannesson et al., 1989). Note there is not necessarily a simple relationship between $\tau$ and glacier size (e.g., Raper and Braithwaite, 2009; Bach et al., 2018). For example, cirque glaciers can be thick and have termini that extend only a little past the equilibrium line altitude, giving them $\tau$s of many decades (e.g., Barth et al., 2018). Conversely, glaciers sourced from large accumulation areas may have termini well below the equilibrium line altitude, and if terminating on steep slopes, also be relatively thin, giving them short $\tau$s of a decade or less (e.g., Franz Joseph N.Z., Purdie et al., 2014).

The three-stage model accurately emulates the autocorrelation function and power spectrum of numerical flowline models of ice dynamics (Christian et al., 2018; Roe and Baker, 2014) and has been shown to produce realistic length responses to climate trends and variability given appropriate choices of $\tau$ and $\beta$ (e.g., Herla et al., 2017).

In the simulations we present here, we did not attempt to assign realistic values of $\tau$ and $\beta$ to each glacier. Rather, for each glacier, we performed three separate simulations with $\tau = 10$, 30, and 100 years, and $\beta = 150$. This approach has two main advantages. First, the subset of glaciers analyzed in Fig. 1 are located in regions with a wide range of glacier sizes. Simulating a range of response timescales thus gives a more complete picture of glacier variability in these regions. Second, it allows us to perform a controlled test of how SNR varies with response timescale, which we will demonstrate has important implications for how geologic records of past glacier variability ought to be best interpreted. This range of $\tau$ encompasses most alpine glaciers (e.g., Haeberli and Hoelzle, 1995; Lüthi and Bauder, 2010; Roe et al., 2017). The value of $\beta$ is a function of glacier geometry: $\beta = A/wH$, where $A$ is the surface area, and $w$ and $H$ are the characteristic width and thickness of the glacier in the terminus zone. Our choice of $\beta = 150$ is based on Hintereisferner in the Austrian Alps ($A = 10$ km$^2$, $w = 400$ m, $H = 170$ m, Herla et al., 2017). The equation for $L'$ is linear, and so the value of $\beta$ does not affect the relative importance of different kinds of climate forcing.

Figure 2 shows the glacier length anomalies ($L'$) that result from the time series of $b'$ (Fig. 1c), and for each of the three $\tau$s considered. In each panel, the heavy line represents the ensemble mean, while the light colors represent each of the 13 ensemble members. Note that the magnitude of the length fluctuations is somewhat arbitrary and will change with catchment geometry.

Comparing the columns of Fig. 2, we see the time series of $L'$ become increasingly smooth as $\tau$ increases. During the pre-industrial period, positive trends in $L'$ across the Northern Hemisphere reflect a cooling trend within the LME (i.e., Fig. 1b). Conversely, during the past century of anthropogenic warming, negative trends in $L'$ are found at all locations for $\tau = 10$ years, but only in the Southern Hemisphere for $\tau = 100$ years. The lack of 20th-century retreat in the Northern Hemisphere partially reflects the muted warming trend in the LME relative to observations (Fig. A1). However, the difference in recent

trends between $\tau = 10$ years and $\tau = 30$ years also illustrates a more general point: namely, that higher $\tau$ values imply a more delayed response to a given climate perturbation. We discuss the implications of this result for future changes in glacier length in Section 4.

In addition to the ensemble-mean forced response, the panels in Fig. 2 also show glaciers exhibit significant differences in ensemble spread. These differences are most clearly illustrated by the SNR values, which reveal two key dependencies. First, at all locations, SNR increases monotonically with increasing $\tau$ (i.e., from left to right in Fig. 2). In the Appendix, we show that SNR also increases with $\tau$ at other locations as well (Fig. A2), suggesting that it is a general property of the climate system. Large-$\tau$ glaciers are usually more reliable indicators of forced climate change than small-$\tau$ glaciers, which are more prone to natural fluctuations. There is a trade-off, however: the larger-$\tau$ glaciers smooth over longer periods and so the temporal resolution of the forced change is degraded.

Second, SNR is substantially greater in the spatially-averaged time series of glacier length (Fig. 2, bottom row) than it is for any individual glacier, similar to what we found with annual mass balance (Fig. 1). This suggests that glacier variability is more likely to reflect external forcing if it is coherent across a large spatial domain.

We address each of these behaviors in the next section, in which we also extend our analysis to the larger set of 76 glaciers, for which variability in summer and winter mass balance were also provided by Medwedeff and Roe (2017, Table A1).

## 3.1 Dependence of signal-to-noise ratio on glacier response time

The dependence of SNR on $\tau$ can be understood by considering the spectra of climate variability and glacier length, which are illustrated schematically in Fig. 3. The spectrum of a time series characterizes the variance as a function of frequency (e.g., Yiou et al., 1996). A climate with no interannual persistence (i.e., no memory) is one that has no dependence on previous years, such that each year's temperature and precipitation are drawn independently from random probability distributions. Such a climate is characterized by a white spectrum—one that has equal variance at all frequencies (Fig. 3, solid blue line). A climate that has persistence exhibits a red spectrum, with more variance at low frequencies (Fig. 3, solid red line). Different climate variables can exhibit different degrees of persistence, with temperature typically showing more persistence (and thus a redder spectrum) than precipitation. Glacier dynamics then act as a low-pass filter of this climate variability, causing the variance of glacier length fluctuations to drop off sharply at frequencies higher than the cut-off frequency of $(2\pi\tau)^{-1}$. As a result, the spectrum of glacier length is generally redder than the unfiltered spectra of temperature and precipitation (Fig. 3 dashed lines; Roe and Baker, 2014).

The spectra of natural and forced variability are shown in Fig. 4, and can be used to understand the variation of SNR with $\tau$ (Fig. 2). They were calculated for the pre-industrial period (1000 to 1880). The rows in Fig. 4 show spectra for annual mass-balance ($b$), winter precipitation ($P$), and summer temperature ($T$). Gray lines represent the spectra at the locations of all 76 glaciers in our data set; the colored lines represent the subset of three glaciers analyzed in Figs. 1 and 2; and finally, the black line represents the average of all individual spectra across the 76 glaciers. Average slopes of the various spectra were calculated using linear regression, and are shown in the bottom left of each panel.

Comparing the spectra of unforced and forced mass-balance variability in the top row of Fig. 4, we find that the unforced spectrum is essentially white (i.e., a flat spectral slope), while the forced spectrum is red (i.e., greater variance at low frequencies). Therefore, when low-pass filtered by the glacier response, more of the forced variability is retained by the glacier, resulting in higher SNR values in glacier length relative to annual mass balance. This effect becomes larger as $\tau$ increases and the filter cutoff shifts to lower frequencies, thus explaining the positive correlation between SNR and $\tau$ found in Fig. 2.

The reason for the different slopes in the mass-balance spectra is due to differences in summer temperature. First, consider the precipitation spectra for both forced and unforced variability (Fig. 4c-d). They are essentially flat at all frequencies, consistent with white noise of internal atmospheric variability. Now consider the temperature spectra of the unforced variability (Fig. 4e). It is only weakly red, with an average slope of $-0.17$; this slope equates to the exponent $\alpha$ in a spectral power law of the form, $P \propto f^{\alpha}$. Finally, consider the temperature spectra for forced variability (Fig. 4f). These are much more strongly red, with an average slope of $-0.61$, and are thus clearly responsible for the similarly reddened spectra of the forced mass-balance (Fig. 4b). While there are some variations among the 76 glacier locations (grey lines in Fig. 4), the broad features of these spectra are seen throughout our network.

Interestingly, the redness of the forced mass-balance spectra in Fig. 4 is not evident in the spectrum of volcanic forcing, which is white at sub-decadal frequencies (Fig. A3). It must therefore be caused by reddening mechanisms inherent to the climate system, such as ocean heat storage and sea-ice feedbacks (Stenchikov et al., 2009; Miller et al., 2012). GCM simulations have further shown that strong, abrupt forcing from volcanic aerosols can also be reddened by enhanced heat exchange with the deep ocean, due to a strengthening of both vertical mixing and the Atlantic Meridional Overturning Circulation (AMOC) (Stenchikov et al., 2009). This unique response to volcanic forcing may help explain why the forced spectra of $T$ and $b$ are so much redder than the unforced spectra (Fig. 4). However, further research is needed to fully understand the difference in spectral slopes between forced and internal temperature variability.

## 3.2 Dependence of signal-to-noise ratio on spatial scale

We now evaluate the spatial coherence of the forced and unforced responses among the set of 76 glaciers. Our 76 glaciers are clustered into a few specific glacierized regions, which allows us to evaluate coherence within, and among, such regions. To facilitate comparison, we fix $\tau = 10$ yr, and calculate the length fluctuations using Eq. 7 at each of the 76 locations for i) full mass-balance, ii) temperature-dependent mass balance, and iii) precipitation-dependent mass balance. Taking each of these three cases in turn, we calculate the correlation coefficient of the length history of each glacier with that of every other glacier, for the pre-industrial era (1000 to 1880). We present the results in matrix form in Figure 5, for cases: (i) top row, (ii) middle row, and (iii) bottom row. Although we show only the $\tau = 10$ yr, results for $\tau = 30$ and $100$ yr were similar.

For natural variability (Fig. 5a), we find strong correlations at the scale of individual glacierized regions, but not at larger scales. For example, while all glaciers in the Alps are strongly correlated with each other, they are not significantly correlated with glaciers in Scandinavia. This implies that internal variability has a relatively short decorrelation length scale, consistent with synoptic-scale atmospheric variability (e.g., Wallace and Hobbs, 2006). When precipitation variability alone is considered (Fig. 5c) we see anticorrelations between the Alps and Scandinavia, and within the set of Pacific Northwest glaciers. These

anticorrelations reflect dipoles in the patterns of interannual variability due to latitudinal shifts in the storm tracks associated with the North Atlantic Oscillation and the Pacific North American patterns, and have been documented previously for glacier mass balance (e.g., Bitz and Battisti, 1999; Bonan et al, 2019). Such anticorrelations are not generally strong enough to dominate the coherence of the overall length variability (Fig. 5a)

For forced climate variability, cross-correlations among glacier length are significantly positive among a large majority of glacier pairs (92%; Fig. 5b), indicating that climate changes resulting from external forcing are globally coherent (Fig. 5b). Moreover, when multiple time series of glacier length are averaged across different mountain ranges, the incoherent internal noise is damped relative to the coherent forced signal. This explains why the SNR of the averaged time series of glacier length is greater than the SNR at any individual glacier (Fig. 2). Meanwhile, the decomposition of the forced correlations (Figs. 4d,f) shows that the global coherence of forced glacier variability comes overwhelmingly from $T$ rather than $P$. Precipitation-driven variability is a source of noise that mostly weakens the correlations, but does not change their global coherence.

### 3.3 The relative importance of precipitation versus temperature for mass balance and length fluctuations

The preceding analysis has shown that forced changes in glacier length are driven primarily by globally-coherent changes in summer temperature. However, this result provides little insight into the relative importance of temperature versus precipitation in driving glacier variability more generally. These contributions are quantified in Fig. 6, which shows the ratio of $T$-driven variance to total variance, both in annual mass balance (Fig. 6a) and in glacier length (Fig. 6b, assuming $\tau = 10$ years). Because variability in $T$ and $P$ are not significantly correlated at any glacier location, this ratio approximately represents the fraction of total variance that can be attributed to $T$. The difference between the two panels is shown in Fig. 6c.

In the case of annual mass balance (Fig. 6a), $T$ accounts for more than half the variance at 58 out of 76 glaciers. The relatively few glaciers where precipitation variability plays a larger role are mostly located in maritime environments with large storm-track variability, such as Alaska, the Pacific Northwest, and the Andes (Fig. 6a). In contrast, variability in glacier length is dominated by temperature everywhere (Fig. 6b), including at glaciers (like South Cascade) where precipitation accounts for most of the variability in annual mass balance. Averaged across all glaciers, temperature accounts for 67% of the total variance in annual mass balance, and 83% of the total variance in glacier length when $\tau = 10$ years. Temperature's share of the variance continues to increase with increasing $\tau$, but more modestly (to 86% when $\tau = 30$ years and 89% when $\tau = 100$ years).

Why does temperature variability exert a greater influence on glacier length than on annual mass balance? Recall from Fig. 4 that the spectrum of $T$ is redder than the spectrum of $P$. This difference is especially pronounced in the forced time series, but it is also evident in the unforced time series. Glaciers filter out variance at the highest frequencies, where precipitation's contribution to annual-mass-balance variance is greatest. Thus, for the same reasons that SNR is enhanced by a glacier's filtering properties, the contribution of temperature to glacier-length variability is enhanced relative to that of precipitation. This means that variability in glacier length primarily reflects low-frequency variability in summer temperature, even where mass-balance variability is more strongly influenced by winter precipitation.

## 3.4 Roles of individual forcings

Finally, we evaluate the relative importance of the different climate-forcing factors in these simulations. In addition to the full 13-member ensemble, the LME archive also contains smaller ensembles of simulations representing the climate response to single factors. The factors (and the number of ensemble members) are: greenhouse gases (3), volcanic aerosols (5), industrial aerosols (5), and changes in solar and orbital patterns (3). As in the full-forcing ensemble, we use the ensemble mean to approximate the climate response to each individual factor. However, it is important to note that these approximations contain substantially more noise than the full ensemble mean because there are fewer ensemble members.

Figure 7 shows the time series of glacier length anomalies induced by each individual forcing factor, averaged among all ensemble members and all glaciers in the Northern Hemisphere (left) and Southern Hemisphere (right). Results are presented for $\tau = 10$ years (top) and $\tau = 30$ years (bottom). The contributions from the solar and orbital forcing were negligible and are not shown.

During the pre-industrial era, most of the forced variability in glacier length can be attributed to volcanic aerosols, as indicated by the similar behavior of the red and green lines in Fig. 7. Over the last century, however, anthropogenic factors have played the largest role. In the Southern Hemisphere, the full-forcing time series closely follows greenhouse-gas-driven trends beginning around the year 1850. In the Northern Hemisphere, by contrast, retreat due to greenhouse gases is largely offset by industrial aerosol emissions during the modern era. While it is well known that industrial aerosols provided radiative cooling over the 20th century, warming trends over this period are generally lower in the LME than in observations (Fig. A1), suggesting that the model's aerosol forcing may be too strong, or that its transient climate sensitivity may be too low. Whatever the cause, the suppressed 20th-century warming trend in many regions within the LME explains why, in some locations, our simulations appear to show less glacier retreat than has been observed.

## 4 Summary and Discussion

In this study, we have combined an ensemble of numerical climate model simulations and a glacier length model to evaluate the relative importance of climate forcing and internal climate variability in driving glacier-length fluctuations over the last millennium. While the potential importance of internal variability has been noted before, this is the first study to evaluate the relative importance of natural forcing vs. internal variability in the pre-industrial era, and the accompanying spatial patterns of glacier response. We estimated annual mass-balance anomalies at 76 glaciers around the world using simulated time series of summer temperature and winter precipitation from the 13-member CESM Last Millennium Ensemble. Using the three-stage linear model of Roe and Baker (2014), we then converted these mass-balance anomalies into glacier-length anomalies for a range of glacier response timescales, thus capturing the diversity of behavior exhibited by glaciers of different sizes and geometries. Because the ensemble simulations differ only in their initial conditions, the responses of mass balance and glacier length to external radiative forcing are mostly captured by the ensemble mean, while internal unforced variability is represented by departures from the ensemble mean (i.e., the ensemble spread). The ratio of ensemble-mean variance to ensemble-spread variance is defined as the signal-to-noise ratio (SNR), and represents the fraction of total variance driven by external forcing.

While SNR varies by location, we found that two factors influence SNR more generally. First, SNR is greater for glacier length than for annual mass balance, and continues increasing with increasing glacier response timescale, $\tau$. This timescale-dependence reflects differences in how forced and unforced mass-balance variance is distributed across the frequency spectrum. Specifically, because forced variance has a redder spectrum than unforced variance, the forced signal is amplified relative to unforced noise when low-pass-filtered by a glacier. This explains why a glacier like South Cascade, which does not exhibit a significant forced response in annual mass balance due to high interannual variability (Fig. 1, red), nonetheless shows a significant forced response in glacier length in our simulations (Fig. 2, red).

Second, for both mass balance and glacier length, SNR is enhanced by averaging multiple length histories from different locations. This is because forced changes in mass balance tend to be globally coherent, reflecting a global-scale temperature response, while unforced variability has little coherence beyond the scale of one glacierized region. Our analyses suggest that forced changes in glacier length are driven primarily by globally coherent changes in summer temperature; and that, in the pre-industrial era, those arise from volcanic-aerosol forcing.

Our results have implications for how past glacier variability should be interpreted. For example, because SNR increases with spatial averaging, a change in glacier length is more likely to be forced when coherent changes are also observed in other regions.

However, it is important to recognize that the amplified signal in large-$\tau$ glaciers comes at the expense of decreased temporal resolution. Thus, while large-$\tau$ glaciers may be more reliable indicators of climate change on centennial timescales, they may entirely miss climate changes that occur on multi-decadal timescales.

Differences in response time also have implications for how glaciers have responded and will respond to global warming. Because glaciers are lagging indicators of climate change, recent glacier retreat has been more modest than the decrease in annual mass balance (Figs. 1 and 2), implying that further retreat is locked in, even in the absence of additional warming (Christian et al., 2018). This disequilibrium is especially pronounced for large-$\tau$ glaciers, whose slow response means that they have only begun to feel the effects of the past century of warming. In future decades, therefore, we expect that large-$\tau$ glaciers will experience the greatest retreat, as they integrate the effects of both past and future warming.

Finally, we have analyzed time series of glacier-length fluctuations. Of course, in actuality glacial history is recorded mostly by the dating of moraines on the landscape. Moraines are created by many different physical processes (e.g., Anderson et al., 2014, for a review of processes and formation times), and dating resolution and accuracy limits the ability to evaluate the coherence of glacier advances in different regions (Schaefer et al., 2009; Balco, 2009). Moreover, different glacier response times affect the phase relationship between the timing of a cooling and the response of glacier length. So a careful assessment of glacier dynamics should be incorporated into any comparison of moraine histories from different locations, as done, for example, in Young et al. (2011). A natural extension of the present work would be to incorporate a simple moraine model (e.g., Gibbons et al., 1984; Anderson et al., 2014), and to evaluate the spatial coherence of moraine statistics.

Our analyses have been made possible by ensemble climate modeling. As done here, such ensembles can be used to decompose the contributions due to internal variability and natural and anthropogenic forcing. However, we have used only one climate model. As other ensembles become more widely available it will be important to evaluate different climate models,

and also different estimates of the natural and anthropogenic forcing, for which there are still significant uncertainties (Schmidt et al., 2011).

*Data availability.* Output from the CESM Last Millennium Ensemble can be downloaded from the National Center for Atmospheric Research (NCAR) Climate Data Gateway: https://www.earthsystemgrid.org/.

## Appendix A: Statistical significance of SNR

We consider a signal-to-noise ratio (SNR) to be statistically significant if we can reject the null hypothesis that $\text{SNR} = 0$ with at least 99% confidence ($p < 0.01$). To determine the threshold for statistical significance, we performed a Monte Carlo

simulation consisting of 100,000 ensembles of 13 randomly generated Gaussian time series (i.e., noise). We set the length of each synthetic time series based on the estimated degrees of freedom in the given climate variable, as described below. For each 13-member ensemble, we compute the SNR as in Eq. 6. This yields a set of 100,000 synthetic SNR values which have a mean of 0.083 and a distribution that depends on the number of degrees of freedom in the time series. Because the true SNR of each ensemble is equal to 0, we set the threshold for statistical significance to be the 99th percentile of SNR values within

the distribution.

At all glacier locations within the LME, we find that time series of winter precipitation and summer temperature exhibit an $e$-folding decorrelation scale of no more than 1 year. Thus, for time series of summer and winter mass balance, we assume the number of degrees of freedom is equal to half the number of years in the stated time period Leith (1973). This yields significance thresholds of 0.098 for the pre-industrial and 0.125 for the modern.

For the glacier length time series, we find that the e-folding decorrelation scale is on average equal to about $1.5\tau$. Over the pre-industrial period, this implies about 30, 10, and 3 degrees of freedom for $\tau = 10$, 30, and 100 years, respectively. The corresponding SNR significance thresholds are found to be 0.15 for $\tau = 10$ years, 0.22 for $\tau = 30$ years, and 0.49 for $\tau = 100$ years. All of our simulated glacier length time series have SNR values that exceed these thresholds.

*Author contributions.* Huston and Siler performed the analysis and wrote the first draft. Roe and Siler designed the study. All authors helped

interpret the results and edit the paper.

*Competing interests.* The authors have no competing interests.

*Acknowledgements.* We thank John Fasullo for providing time series of individual forcing agents used in the LME. We thank the Oregon State University Honors College for supporting this research. This work was funded by the US National Science Foundation grants AGS-2024212, AGS-1805490, AGS-1903465. LDEO contribution number XXXX.

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

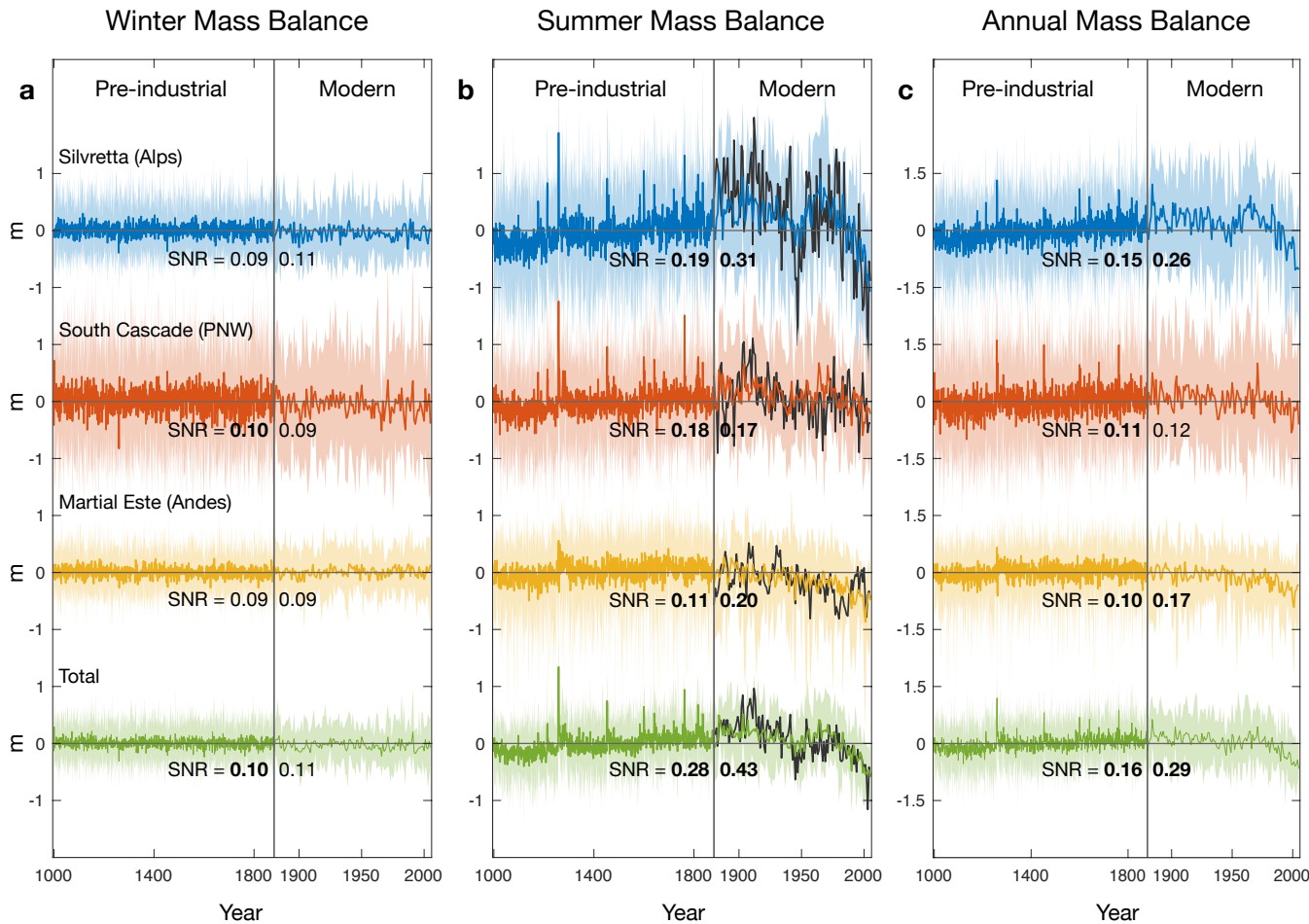

**Figure 1.** One thousand years of mass-balance anomalies modeled using the LME output, for three different locations. (a) winter mass balance, (b) summer mass balance, and (c) annual mass balance in the Alps (blue), the Pacific Northwest (red), and the Patagonian Andes (yellow), along with their average (green). Heavy colors represent the ensemble mean, while lighter colors represent individual ensemble members. The vertical grey line at the year 1880 marks the approximate transition from the pre-industrial era to the modern era, after which the time axis on each panel is dilated by a factor of 4 to enhance visibility. To evaluate the accuracy of LME temperatures over the modern era, we also plot values of $b'_s$ calculated from the observational NASA-GISS surface temperature data set in black. SNR values are given for each glacier and era, with bold font indicating statistical significance (99% confidence; see Appendix).

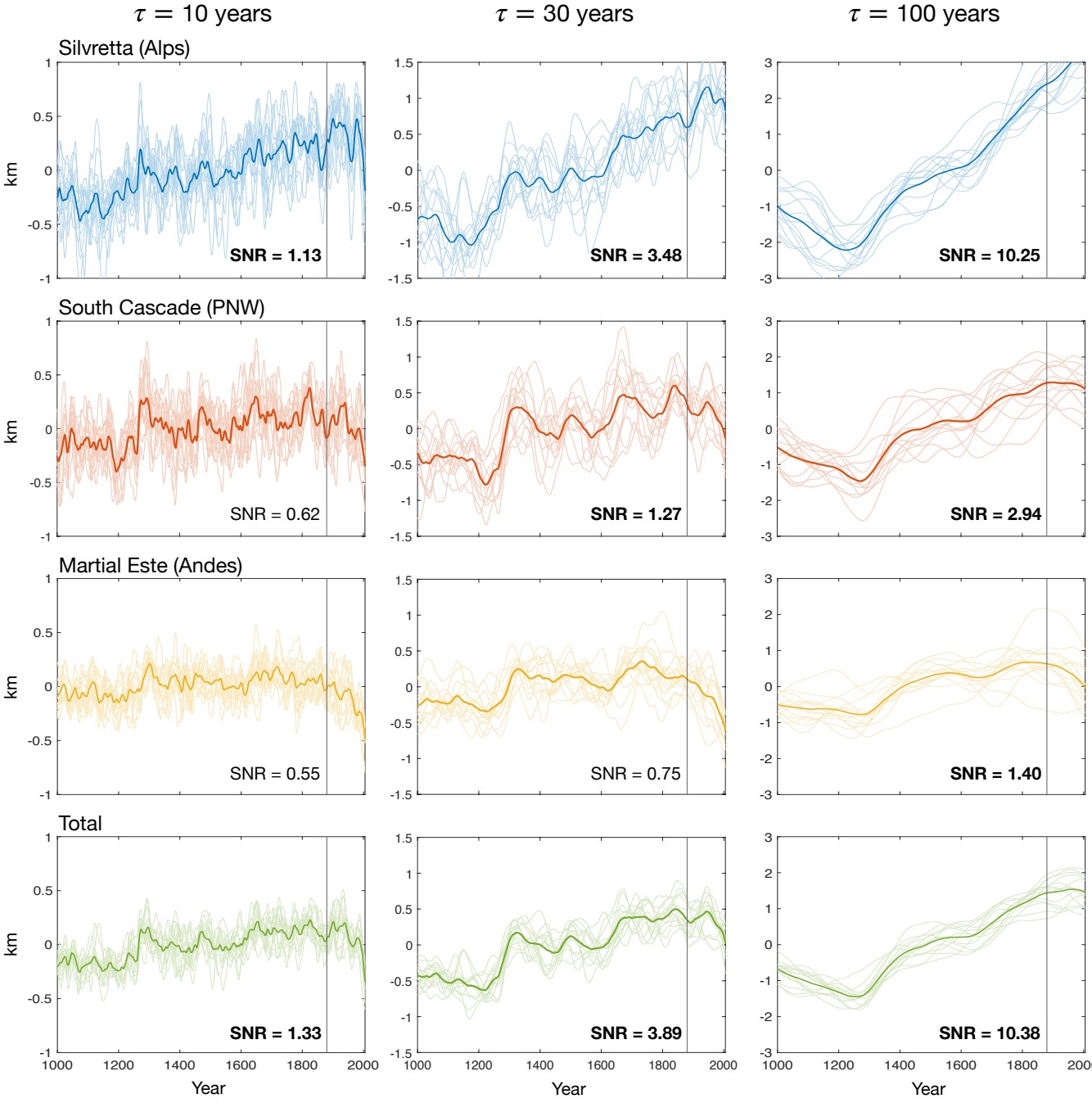

**Figure 2.** Simulated glacier-length anomalies for representative glaciers in the Alps (top row), the Pacific Northwest (second row), and the Patagonian Andes (third row). The fourth row shows the sum of all three glaciers. Each column shows results for different glacier response times: $\tau = 10$ years (left), $\tau = 30$ years (middle), and $\tau = 100$ years (right). Heavy colors represent the ensemble mean, while shaded colors represent individual ensemble members. The vertical grey line at the year 1880 marks the transition from the pre-industrial era to the modern era. SNR values are computed over the pre-industrial era only. Bold font indicates $\mathrm{SNR} > 1$, meaning that the forced signal exceeds the noise of internal variability.

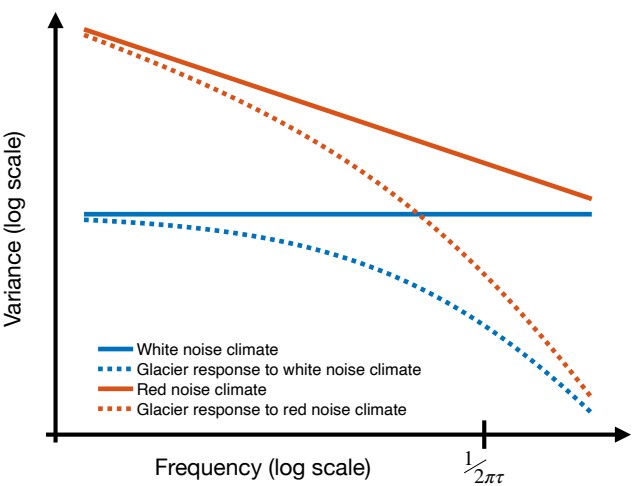

**Figure 3.** Schematic illustration of time series power spectra, representing the relative magnitude of variance as a function of frequency. A white spectrum is flat (solid blue line), indicating uniform variance at all frequencies. A red spectrum has a negative slope (solid red line), indicating greater variance at low frequencies. A glacier acts as a low-pass filter, resulting in a redder spectrum for glacier length than for annual mass balance, which varies with temperature and precipitation. The variance in annual mass balance that is retained in time series of glacier length drops sharply beyond the cut-off frequency of $(2\pi\tau)^{-1}$.

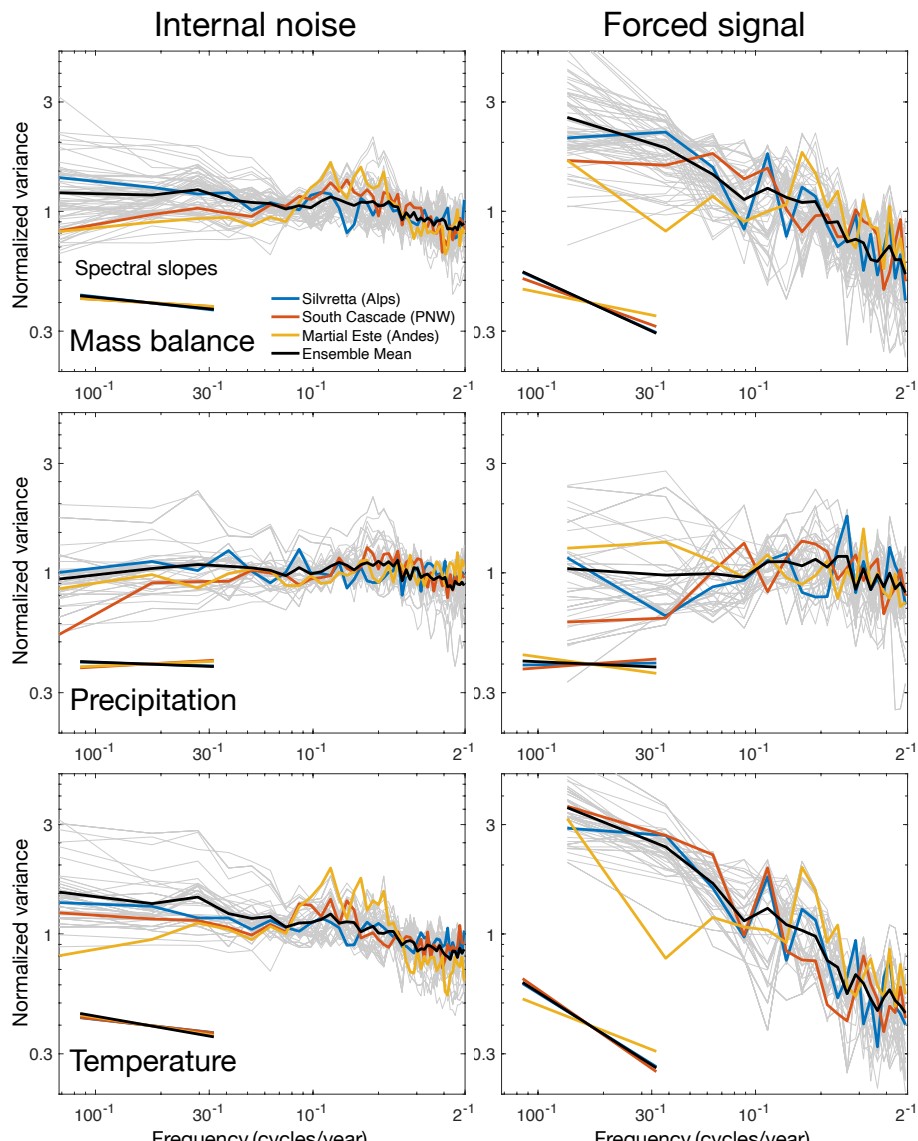

**Figure 4.** Power spectra of forced and unforced variability for the pre-industrial era (1000 to 1880). Internal (unforced) variability is shown in the left column and forced variability in the right column. Top row is annual mass balance, middle row is winter precipitation, and bottom row is summer temperature. Spectra are shown for Silvretta (blue), South Cascade (red), Martial Este (yellow), and the other 73 glaciers (gray). The black line shows the average of all 76 individual spectra. Lines in the bottom left of each panel show the slope of the least-squares regression line for each spectrum. Spectra were computed using periodograms with a Hanning window equal to one-third the length of the data. Unforced spectra were averaged among the 13 ensemble members. Forced spectra were band-averaged to reduce noise, explaining the coarser spectral resolution.

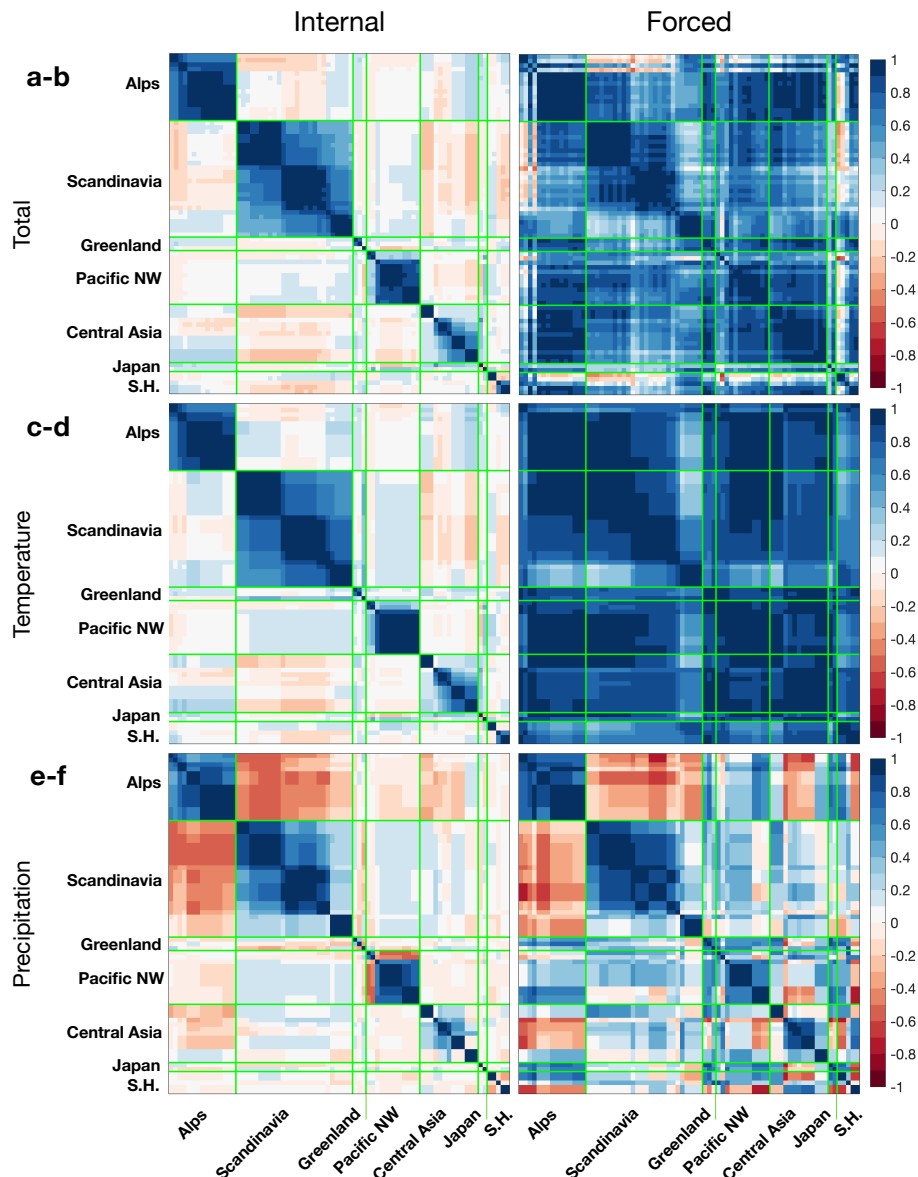

**Figure 5.** Matrix of cross correlations among the network of 76 glacier locations (Table A1), grouped by region. Each column or row in the matrix shows the correlation coefficients of glacier length at one location with glacier length at every other location in the network, calculated for the pre-industrial era (1000 to 1880). Left panels shows results for internal variability, and the right panels shows results for forced variability. Top row shows the results for the full mass-balance variability, while the middle and bottom rows show results for, respectively, the temperature-only and precipitation-only components of the mass balance. Blue colors indicate positive correlations, and red colors indicate negative correlations. A response time of $\tau = 10$ years was used for these calculations.

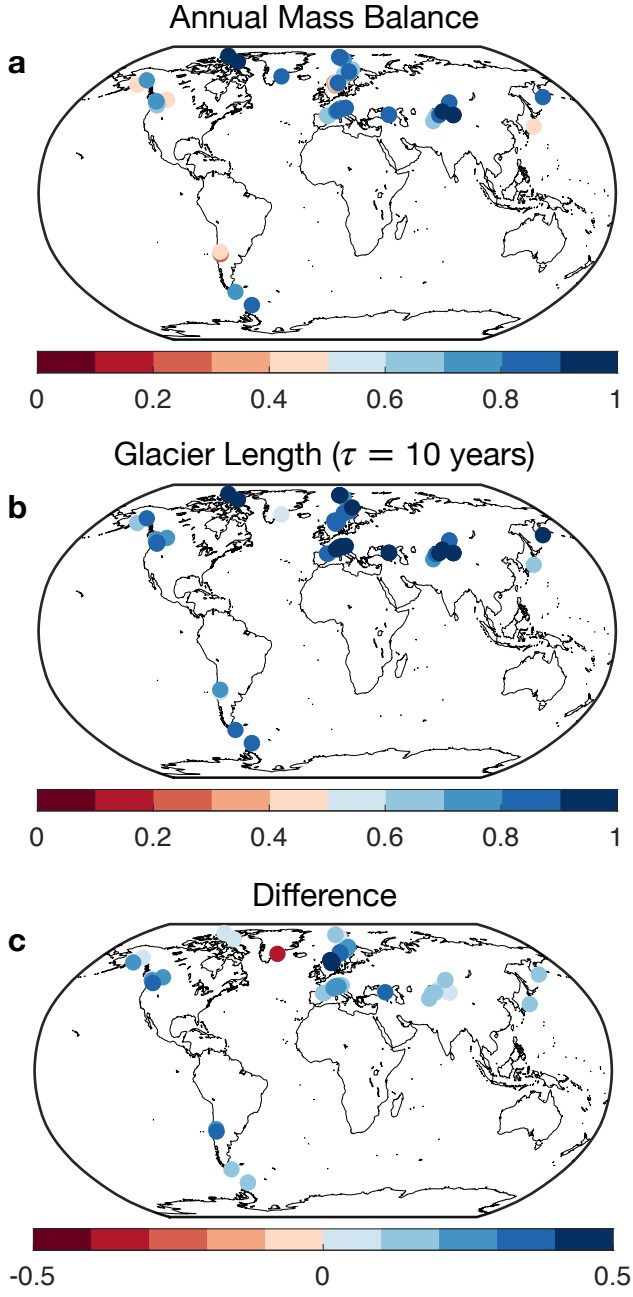

**Figure 6.** The importance of summertime temperature in driving overall variance in mass balance and glacier length across the glacier network, calculated for the pre-industrial era (1000 to 1880). Panel (a) shows the fraction of total variance in mass balance that is due to summer temperature. Panel(b) shows the same thing, but for glacier length. The difference is shown in panel (c), from which we see that, in all but one case (Mittivakkat in Greenland), summer temperature accounts for a larger fraction of variance in glacier length than in annual mass balance. This is due to the low-pass-filtering properties of glacier dynamics.

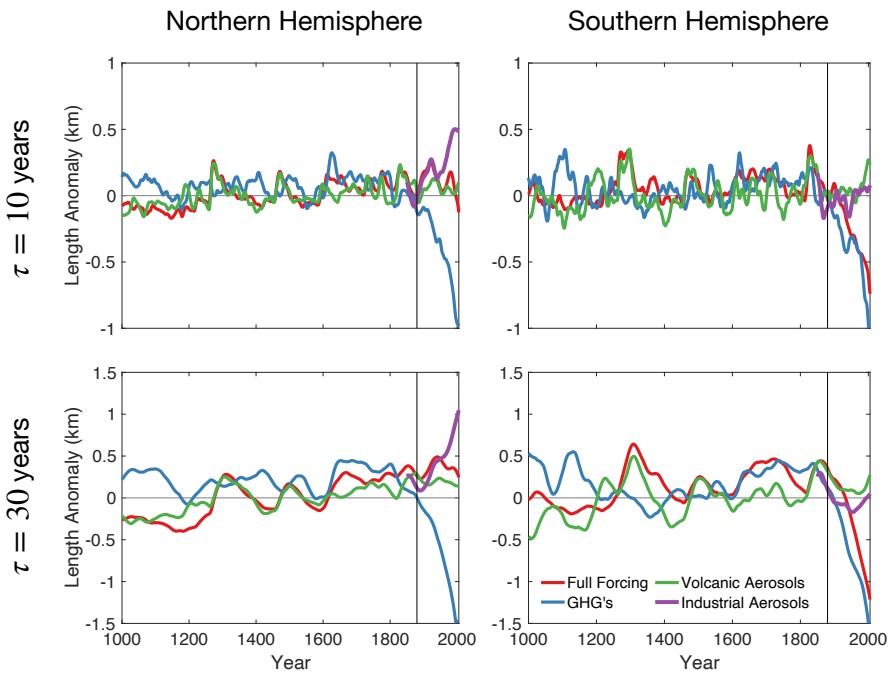

**Figure 7.** Contributions of individual forcing factors to glacier-length variability over the last millennium. Results are shown for response timescales of $\tau = 10$ years (top row) and $\tau = 30$ years (bottom row). The left column shows the average of all Northern-Hemisphere glaciers in our network, and the right column shows the average of all Southern-Hemisphere glaciers. Colors represent the contributions from individual forcing factors, which we approximate from the ensemble-mean time series of summer temperature and winter precipitation within each single-forcing ensemble. The individual forcing factors are greenhouses gases (blue), volcanic aerosols (green), and industrial aerosols (purple). The red line shows the combined influence of all forcing factors diagnosed from the full 13-member ensemble. The gray vertical line marks the transition from the pre-industrial to modern eras at year 1880.

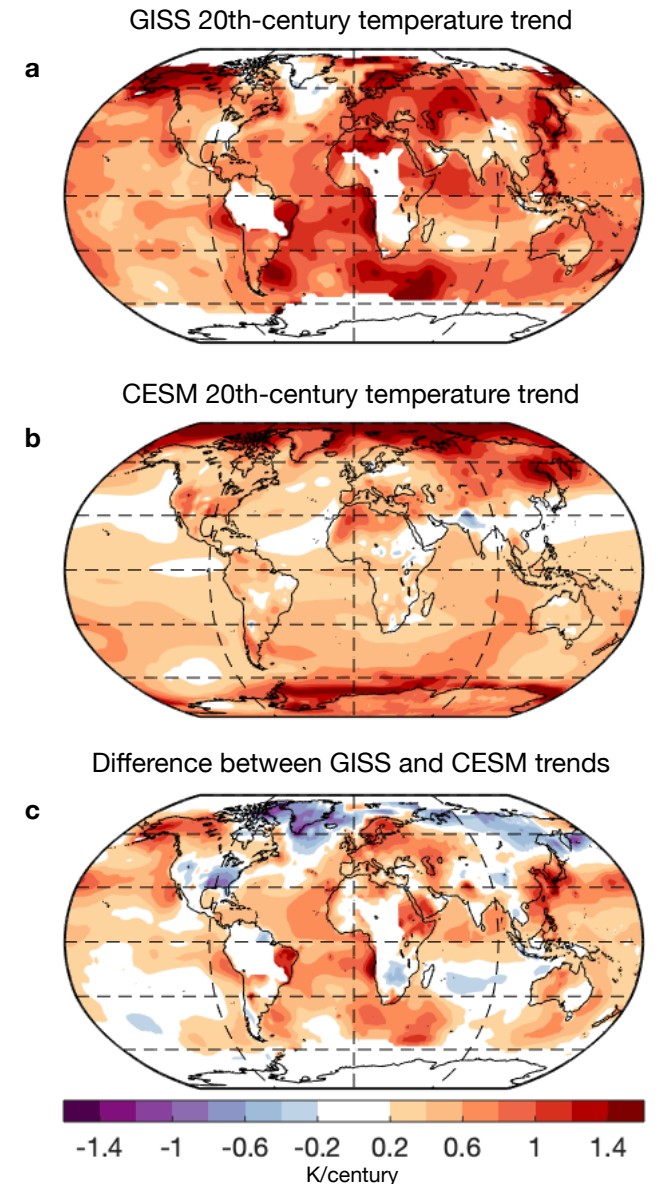

**Figure A1.** Linear trend in 2-m air temperature (in units of K/century) between 1900-1999 in (a) the NASA GISS observational dataset and (b) the CESM LME ensemble mean. (c) The difference between (a) and (b). Warmer temperatures are shown in red, with cooler temperatures shown in blue.

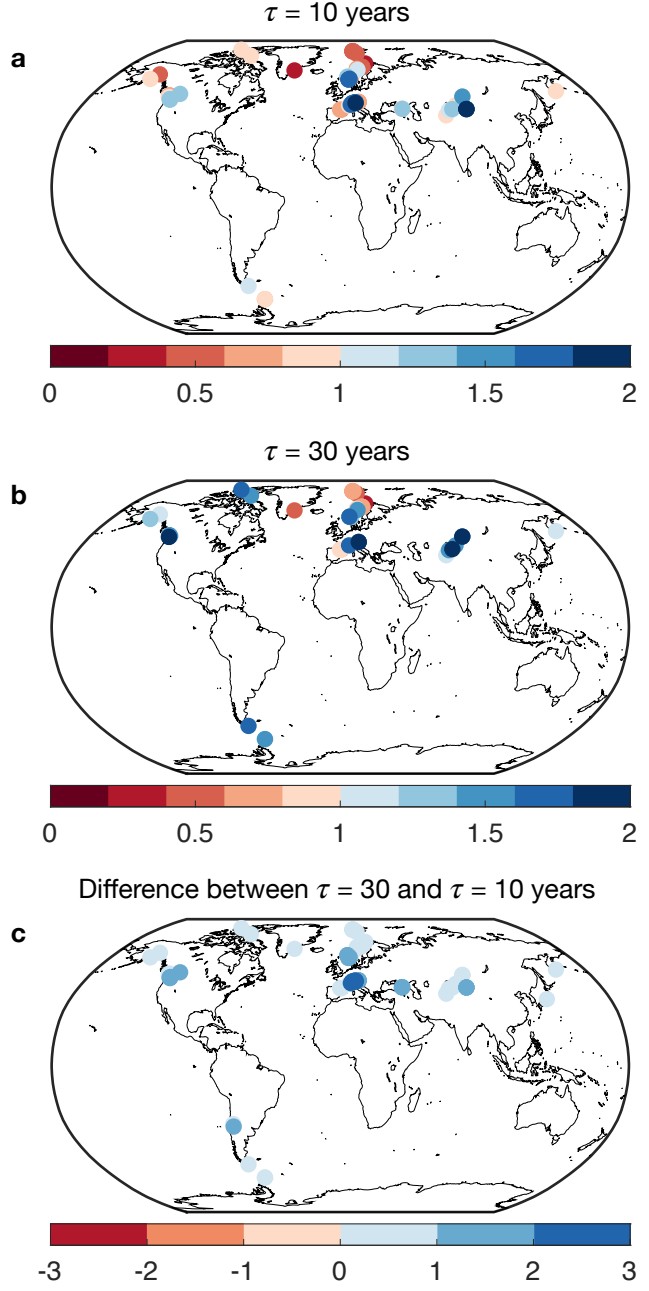

**Figure A2.** The signal-to-noise ratio (SNR) of glacier length for all 76 glacier locations given (a) $\tau = 10$ years and (b) $\tau = 30$ years. (c) The difference between (b) and (a). Red colors in (a) and (b) indicate a larger role for internal variability than forced variability, while blue colors indicate the opposite. Blue colors in (c) indicate a greater role for forced variability when $\tau = 30$ years than when $\tau = 10$ years.

**Figure A3.** Power spectrum of volcanic forcing generated from time series of volcanic aerosol concentrations in the LME (courtesy of John Fasullo).

**Table A1.** Glacier data from Medwedeff and Roe (2017), sorted by region. Glaciers are excluded if observations of summer or winter mass balance span less than 10 years. $\sigma_{b,w}$: the standard deviation of observed winter mass balance (in m). $\sigma_{b,s}$: the standard deviation of observed summer mass balance (in m). $\sigma_P$: the standard deviation of simulated winter precipitation within the LME over the last millennium (1000-1999) (in m). $\sigma_T$: the standard deviation of simulated summer temperature within the LME over the last millennium (1000-1999) (in K). $\alpha$: the ratio of $\sigma_{b,w}/\sigma_P$. $\lambda$: the ratio of $\sigma_{b,s}/\sigma_T$ (in m K$^{-1}$).

| GLACIER/REGION | LAT | LON | $\sigma_{b,w}$ | $\sigma_{b,s}$ | $\sigma_P$ | $\sigma_T$ | $\alpha$ | $\lambda$ |
|---|---|---|---|---|---|---|---|---|
| **ALPS/PYRENEES** | | | | | | | | |
| OSSOUE | 42.77 | -0.14 | 0.54 | 0.78 | 0.07 | 0.69 | 7.49 | 1.14 |
| MALADETA | 42.65 | 0.63 | 0.51 | 0.64 | 0.06 | 0.81 | 7.99 | 0.79 |
| SARENNES | 45.13 | 6.13 | 0.51 | 0.85 | 0.07 | 0.72 | 7.03 | 1.17 |
| GRAND ETRET | 45.47 | 7.21 | 0.58 | 0.47 | 0.09 | 0.76 | 6.55 | 0.61 |
| GRIES | 46.44 | 8.33 | 0.39 | 0.63 | 0.09 | 0.60 | 4.20 | 1.04 |
| CIARDONEY | 45.51 | 7.39 | 0.49 | 1.29 | 0.09 | 0.60 | 5.25 | 2.15 |
| BASODINO | 46.42 | 8.48 | 0.48 | 0.62 | 0.09 | 0.60 | 5.19 | 1.03 |
| SILVRETTA | 46.85 | 10.08 | 0.34 | 0.67 | 0.09 | 0.60 | 3.86 | 1.11 |
| VERNAGT FERNER | 46.88 | 10.82 | 0.22 | 0.42 | 0.09 | 0.60 | 2.52 | 0.70 |
| FONTANA BIANCA / WEISSBRUNNF. | 46.48 | 10.77 | 0.38 | 0.61 | 0.09 | 0.60 | 4.30 | 1.01 |
| JAMTAL F. | 46.87 | 10.17 | 0.25 | 0.49 | 0.09 | 0.60 | 2.88 | 0.81 |
| CARESER | 46.45 | 10.70 | 0.38 | 0.44 | 0.09 | 0.60 | 4.28 | 0.74 |
| WURTEN K. | 47.03 | 13.00 | 0.29 | 0.57 | 0.09 | 0.64 | 3.24 | 0.89 |
| KLEINFLEISS K. | 47.05 | 12.95 | 0.26 | 0.54 | 0.09 | 0.64 | 2.90 | 0.85 |
| GOLDBERG K. | 47.04 | 12.97 | 0.25 | 0.56 | 0.09 | 0.64 | 2.72 | 0.88 |
| **SCANDINAVIA** | | | | | | | | |
| REMBESDALSKAAKA | 60.53 | 7.36 | 0.75 | 0.54 | 0.14 | 0.71 | 5.27 | 0.76 |
| GRAAFJELLSBREA | 60.08 | 6.39 | 0.73 | 0.60 | 0.14 | 0.71 | 5.12 | 0.84 |
| BREIDABLIKKBREA | 60.07 | 6.36 | 0.80 | 0.55 | 0.14 | 0.71 | 5.58 | 0.78 |
| AALFOTBREEN | 61.75 | 5.65 | 1.07 | 0.73 | 0.19 | 0.67 | 5.76 | 1.08 |
| HANSEBREEN | 77.08 | 15.67 | 0.17 | 0.29 | 0.07 | 1.70 | 2.33 | 0.17 |
| STORBREEN | 61.57 | 8.13 | 0.38 | 0.47 | 0.16 | 0.84 | 2.39 | 0.56 |
| HELLSTUGUBREEN | 61.56 | 8.44 | 0.27 | 0.50 | 0.16 | 0.84 | 1.73 | 0.60 |
| NIGARDSBREEN | 61.72 | 7.13 | 0.61 | 0.59 | 0.16 | 0.84 | 3.88 | 0.70 |
| GRAASUBREEN | 61.65 | 8.60 | 0.25 | 0.51 | 0.16 | 0.84 | 1.56 | 0.60 |
| AUSTDALSBREEN | 61.81 | 7.35 | 0.65 | 0.62 | 0.16 | 0.84 | 4.10 | 0.74 |
| OKSTINDBREEN | 66.01 | 14.29 | 0.69 | 0.51 | 0.08 | 1.01 | 8.75 | 0.50 |
| ENGABREEN | 66.65 | 13.85 | 0.78 | 0.71 | 0.12 | 1.00 | 6.21 | 0.70 |
| STORGLOMBREEN | 66.67 | 14.00 | 0.50 | 0.72 | 0.12 | 1.00 | 3.97 | 0.72 |
| TROLLBERGDALSBREEN | 66.71 | 14.44 | 0.53 | 0.57 | 0.12 | 1.00 | 4.21 | 0.57 |

| GLACIER/REGION | LAT | LON | $\sigma_{b,w}$ | $\sigma_{b,s}$ | $\sigma_P$ | $\sigma_T$ | $\alpha$ | $\lambda$ |
|---|---|---|---|---|---|---|---|---|
| **SCANDINAVIA (CONT'D)** | | | | | | | | |
| STORGLACIAEREN | 67.90 | 18.57 | 0.38 | 0.49 | 0.06 | 1.08 | 6.00 | 0.45 |
| RIUKOJIETNA | 68.08 | 18.08 | 0.39 | 0.59 | 0.06 | 1.08 | 6.17 | 0.54 |
| RABOTS GLACIAER | 67.91 | 18.50 | 0.38 | 0.47 | 0.06 | 1.08 | 6.13 | 0.44 |
| TARFALAGLACIAEREN | 67.93 | 18.65 | 0.44 | 0.74 | 0.06 | 1.08 | 7.01 | 0.68 |
| MARMAGLACIAEREN | 68.83 | 18.67 | 0.21 | 0.46 | 0.09 | 1.03 | 2.36 | 0.44 |
| STORSTEINSFJELLBREEN | 68.22 | 17.92 | 0.35 | 0.29 | 0.09 | 1.03 | 3.96 | 0.28 |
| LANGFJORDJOEKULEN | 70.12 | 21.73 | 0.36 | 0.52 | 0.06 | 1.11 | 5.70 | 0.47 |
| HANSBREEN | 77.08 | 15.67 | 0.17 | 0.29 | 0.07 | 1.70 | 2.33 | 0.17 |
| AUSTRE BROEGGERBREEN | 78.88 | 11.83 | 0.12 | 0.30 | 0.05 | 1.48 | 2.36 | 0.20 |
| MIDTRE LOVENBREEN | 78.88 | 12.07 | 0.13 | 0.29 | 0.05 | 1.48 | 2.65 | 0.20 |
| KONGSVEGEN | 78.80 | 12.98 | 0.15 | 0.31 | 0.05 | 1.48 | 3.03 | 0.21 |
| WALDEMARBREEN | 78.67 | 12.00 | 0.13 | 0.17 | 0.05 | 1.48 | 2.66 | 0.11 |
| **GREENLAND/ NE CANADA** | | | | | | | | |
| MEIGHEN ICE CAP | 79.95 | -99.13 | 0.03 | 0.28 | 0.02 | 1.13 | 1.93 | 0.25 |
| DEVON ICE CAP NW | 75.42 | -83.25 | 0.02 | 0.16 | 0.02 | 1.32 | 1.30 | 0.12 |
| MITTIVAKKAT | 65.67 | -37.83 | 0.14 | 0.37 | 0.21 | 1.27 | 0.69 | 0.29 |
| **ALASKA/PNW** | | | | | | | | |
| GULKANA | 63.25 | -145.42 | 0.29 | 0.51 | 0.05 | 1.06 | 6.52 | 0.48 |
| WOLVERINE | 60.40 | -148.92 | 0.80 | 0.65 | 0.12 | 1.04 | 6.91 | 0.63 |
| PEYTO | 51.67 | -116.53 | 0.40 | 0.38 | 0.05 | 0.73 | 7.83 | 0.53 |
| PLACE | 50.43 | -122.60 | 0.31 | 0.47 | 0.19 | 0.88 | 1.60 | 0.54 |
| SENTINEL | 49.90 | -122.98 | 0.67 | 0.47 | 0.19 | 0.88 | 3.43 | 0.54 |
| HELM | 49.97 | -123.00 | 0.41 | 0.56 | 0.19 | 0.88 | 2.11 | 0.64 |
| ZAVISHA | 50.80 | -123.42 | 0.31 | 0.51 | 0.19 | 0.88 | 1.60 | 0.58 |
| SYKORA | 50.87 | -123.58 | 0.27 | 0.54 | 0.19 | 0.88 | 1.40 | 0.62 |
| SILVER | 48.98 | -121.25 | 0.72 | 0.98 | 0.19 | 0.68 | 3.80 | 1.44 |
| NOISY CREEK | 48.67 | -121.53 | 0.99 | 1.13 | 0.19 | 0.68 | 5.22 | 1.66 |
| SOUTH CASCADE | 48.37 | -121.05 | 0.65 | 0.58 | 0.16 | 0.68 | 4.18 | 0.85 |
| NORTH KLAWATTI | 48.57 | -121.12 | 0.91 | 1.15 | 0.16 | 0.68 | 5.83 | 1.69 |
| **CENTRAL ASIA** | | | | | | | | |
| DJANKUAT | 43.20 | 42.77 | 0.41 | 0.46 | 0.07 | 0.74 | 5.93 | 0.62 |
| GARABASHI | 43.30 | 42.47 | 0.17 | 0.41 | 0.07 | 0.74 | 2.48 | 0.56 |
| TBILISA | 43.13 | 42.47 | 0.14 | 0.31 | 0.07 | 0.74 | 2.05 | 0.43 |
| ABRAMOV | 39.63 | 71.6 | 0.32 | 0.41 | 0.08 | 0.93 | 4.02 | 0.44 |

| GLACIER/REGION | LAT | LON | $\sigma_{b,w}$ | $\sigma_{b,s}$ | $\sigma_P$ | $\sigma_T$ | $\alpha$ | $\lambda$ |
|---|---|---|---|---|---|---|---|---|
| **CENTRAL ASIA (CONT'D)** | | | | | | | | |
| GOLUBIN | 42.46 | 74.49 | 0.15 | 0.23 | 0.04 | 0.72 | 3.74 | 0.32 |
| TS.TUYUKSUYSKIY | 43.05 | 77.08 | 0.21 | 0.44 | 0.03 | 0.65 | 7.94 | 0.68 |
| SHUMSKIY | 45.08 | 80.23 | 0.08 | 0.42 | 0.03 | 0.74 | 2.78 | 0.57 |
| URUMQI GLACIER NO. 1 W-BRANCH | 43.08 | 86.82 | 0.09 | 0.44 | 0.01 | 0.58 | 15.08 | 0.76 |
| URUMQI GLACIER NO. 1 | 43.08 | 86.82 | 0.09 | 0.33 | 0.01 | 0.58 | 15.46 | 0.57 |
| URUMQI GLACIER NO. 1 E-BRANCH | 43.08 | 86.82 | 0.06 | 0.36 | 0.01 | 0.58 | 10.18 | 0.62 |
| MALIY AKTRU | 50.08 | 87.75 | 0.27 | 0.30 | 0.05 | 0.69 | 5.76 | 0.43 |
| VODOPADNIY (NO.125) | 50.10 | 87.70 | 0.10 | 0.22 | 0.05 | 0.69 | 2.09 | 0.32 |
| LEVIY AKTRU | 50.08 | 87.72 | 0.15 | 0.24 | 0.05 | 0.69 | 3.16 | 0.35 |
| **JAPAN** | | | | | | | | |
| HAMAGURI YUKI | 36.60 | 137.62 | 1.63 | 1.52 | 0.08 | 0.70 | 19.69 | 2.18 |
| KOZELSKIY | 53.23 | 158.82 | 0.61 | 1.22 | 0.09 | 1.05 | 6.84 | 1.16 |
| **SOUTHERN HEMISPHERE** | | | | | | | | |
| ECHAURREN NORTE | -33.57 | -70.13 | 1.24 | 0.71 | 0.13 | 0.63 | 9.71 | 1.12 |
| PILOTO ESTE | -32.22 | -70.05 | 0.58 | 0.55 | 0.13 | 0.63 | 4.58 | 0.87 |
| MARTIAL ESTE | -54.78 | -68.40 | 0.28 | 0.49 | 0.04 | 0.49 | 7.10 | 1.00 |
| JOHNSONS | -62.66 | -60.35 | 0.17 | 0.22 | 0.05 | 0.99 | 3.38 | 0.22 |
| HURD | -62.68 | -60.40 | 0.14 | 0.30 | 0.05 | 0.99 | 2.72 | 0.31 |