# Peer review of "Understanding Drivers of Glacier Length Variability Over the Last Millennium"

_The Cryosphere, 2020_

## Referee Comment (RC1) · Anonymous Referee #1 · 27 Oct 2020

Review of

"Understanding Drivers of Glacier Length Variability Over the Last Millennium"

This manuscript uses a linear glacier model forced with output from a global climate model to explore controls on glacier mass balance and length variability over the last Millennium. The relative importance of forced versus internal variability is emphasized. The manuscript explores the importance of glacier response time in controlling the resulting glacier length simulations. The regional correlation of glacier length is revealed as an important indicator of past, forced temperature changes. The importance of single forcings (i.e., GH gases, insolation, etc) in glacier length variability is also considered.

Ultimately there are a number of new insights from this work: 1) that internal variability drives length changes in mountain glaciers when response times are less than a few decades; 2) that glaciers with longer response times are more likely to capture the effects of external variability; and 3) that external forcing dominates when glaciers response is averaged across widely separated regions.

**Major comments**

As it stands I find that the analyses are very well done and a lot of new insight is provided. This manuscript provides new theory for interpreting past glacial length changes. It uses an approach that honors understandings of climate variability that have been around in the atmospheric science community for decades but have not yet made their way into the glaciological or glacial geology communities.

With a bit of work to improve legibility, the manuscript can be a long-lasting contribution. Further synthesis of the results is needed. Because the implications of this work are most pertinent for folks working on paleoglaciers and the interpretation of the moraine record, the manuscript should be adjusted to make it more accessible for these folks. If not adjusted the important insights provided here might be missed or overlooked. Further suggestions to aid this effort are provided in the minor comments below.

In some parts of the manuscript so much information is presented that the reader is overwhelmed. This is the case even for a reviewer who is familiar with the methods the author's apply. The analyses vary rapidly in spatial scope (3 glaciers, 76 glaciers, hemispheric) and temporal scope (pre-1880, post-1880) or vary between different components of the mass balance (summer, winter, annual) or the response time (10, 30, 200 yrs). While it is a good 'problem' to have too many analyses I think the manuscript would be more effective if there were a more central thread to follow. The introduction as it reads does not lay out the analyses to come. Removing some of the extraneous analyses and points would also help. The authors could also take a bit more space to explain their analyses/ results if more extraneous points were removed. Better transitions between sections 3.1-3.4 are also needed.

The figures can also be overwhelming as they provide a lot of information. Again this is positive but the authors should consider how these figures could be simplified while still allowing the main points to come across. Perhaps some of the multi-paneled figures could be moved into the Appendix or Supplemental and be replaced with simplified versions in the main text to improve legibility?

Along these lines the authors might consider removing the analysis related to individual forcings (section 3.4) and glacier length and save it for another manuscript. Figure 6 was interesting but I felt it was rather hard to take much away from it. From my view there is more than enough new contributions from Sections 3.1-3.3.

The manuscript could also be streamlined by narrowing the temporal scope. If the authors just analyzed 1000 to 1880 then any issues with the GCM output poorly representing the more recent climate would be removed or even put in the supplemental material. Furthermore removing the 1880 to present time period would also lessen some of the more complex items the authors discuss in the manuscript now, allowing for a more streamlined read.

The text should address the paleoglacier literature more directly. The paper dances around ideas related to the moraine record. I think the authors should meld in a bit of the the analysis of Anderson et al., 2014 into the discussion. Who in their table 2 also show SNR as related to response time. While the Anderson paper uses the one-stage version of the linear model the same general trends are apparent with either the 1-stage or 3-stage. Their analysis if nothing else supports the conclusions of this paper: larger SNR result from longer glacier response times. Authors should also consider referencing Young et al. (2011) who at least note the potential role of varying response time in controlling the timing of moraine formation.

The manuscript lays out some very important conclusions for those interpreting the moraine record/ paleoglacier lengths, but these points are not highlighted as much as they could at the end of the manuscript. Or they are overwhelmed by other points made in the discussion/conclusion section.

**Minor comments**

Line 20-21. A more accessible description of internal climate variability would be helpful here for those that do not think about climate in this way.

Section 1 Introduction would benefit the reader more of a bit more by introducing say glacier response times and signal-to-noise ratios of glacier length/mass balance here. As it stands I was a bit surprised by the breadth of the analyses as I read down.

line 84. Perhaps a bit more about the data used to create these temperature anomalies would be useful here. Really just so the reader doesn't need to go looking.

Line 179-180 See Anderson et al., 2014 Table 2 for similar results.

Line 185. The extension of the analysis to 76 additional glaciers is a surprise. The reader would benefit from a bit of an outline of the simulations at the end of the introduction.

Section 3.1 "Dependence of SNR on timescale" would be more clear as "Response time and SNR" same for Section 3.2 maybe "Spatial scale and SNR"

Line 196. I find this paragraph to be accurate but overwhelming if you are not already an atmospheric/climate scientist or previously familiar with the author's approach to linking climate variability to glacier response. So more simple explanations and simplifying will help.

Line 197. perhaps take a bit more space here to explain what 'white' noise means for those who are unfamiliar.

Line 202. This sentence could be rewritten to just state what the source of the differences in slope are. Right now my brain is a bit overwhelmed looking at all the data in figure 3.

Line 255-257. This is a really important observation one that helps bridge the gap between Quaternary geologists and the authors', more atmospheric science-based approach.
Line 274-276. This seems like a bit of an understatement on the part of the authors. There are scant few glaciers that have advanced through the last century.

Section 4 (Summary and Discussion) would be improved with more synthesis in how these results relate to folks who interpret moraine chronologies and past glacier fluctuations. For example: "The preceding analysis has shown that forced changes in glacier length are driven primarily by globally-coherent changes in summer temperature." from line 239-240 would be good to emphasize here.

Line 299. It would help to put a range of values of this in numbers as right now I can think of mountain ranges that vary in area by orders of magnitude. The 'individual mountain range' phrase was used above as well.

Line 300-302. The discussion from Anderson et al, 2014 is relevant here for moraine ages across the western US for the LGM. It is a real-world example that ties into a similar analysis. Its inclusion would add depth to the discussion.

Line 316. should be a new paragraph.

**Figures**

Fig. 2. Perhaps the effect of glacier response time would be better shown if the y-axes were the same for all panels.

Fig. 3 Labels on one of the panels outlining what is low frequency and what is high frequency. Otherwise the reader needs to do a bit of math in their head if they are not used to looking at such figures.

Perhaps the spectral slopes portion of the panels could each be in a box or subplot within the panel so they are differentiated from all of the other lines in each panel?

The authors should consider de-emphasizing the data (maybe with transparency?) and emphasizing spectral slopes, which I find to be a more clear expression of the main point.

Fig. 4 This figure is again packed with information. While it is quite interesting and there is a clear trend I come away overwhelmed with information.

Fig. 5  caption: "positive values indicate that summer
temperature accounts for a larger fraction of variance in glacier length than in annual mass balance." I think the caption is incorrect here. As written temperature accounts for a larger fraction of variance in annual mass balance than in glacier length.

I found that the text written about Figure 5 in section 3.3 was easier to follow than the figure itself. It might be more effective for the authors to just describe the take home here in the text with a few statistics?

Fig. 6 I think this figure could be improved by defining the lines with different linestyles. Right now the take away from this figure is not clear as well. In panels a and b, the lines are so dense that the reader has to work very hard to differentiate them.  Maybe there is an easier way to present these results with a couple panels that are more legible or in a table with statistics?

References

Anderson, L.S., Roe, G.H., Anderson, R.S., 2014. The effects of interannual climate variability on the moraine record. Geology 42, 55–58. https://doi.org/10.1130/G34791.1
Young, N.E., Briner, J.P., Leonard, E.M., Licciardi, J.M., Lee, K., 2011. Assessing climatic and nonclimatic forcing of Pinedale glaciation and deglaciation in the western United States. Geology 39, 171–174. https://doi.org/10.1130/G31527.1

---

## Referee Comment (RC2) · Anonymous Referee #2 · 7 Nov 2020

This study attempts to quantify the relative roles of internal climate variability vs. external forcing on length fluctuations in glaciers over the last millennium. I believe this to be an excellent and important study, which I would recommend for publication after revisions.

Comments: L25–29, 40–41: As far as I can tell from this discussion, these studies do not directly address glacier length variability, rather temperature variability. Please clarify this. L47–53: Since the results of the paper rely on the fact that the ensemble spread can be used as a proxy for internal climate variability, I would like to see some more discussion of this. Have other studies used the LME for this purpose? How realistic is the magnitude of variability in LME? It would be good to cite some other papers that have similarly used ensembles of climate models for the purpose of disentangling

forced and unforced variability. Eq. 6: Please make clearer exactly how you compute this. The numerator is clear enough, but L90 "total variance across all ensemble members" is not totally clear. L140–144: There are a few studies that have looked at the dependence of response time on size on a global scale; see Raper and Braithwaite (2009), Bach et al. (2018). L142: The reference Barth et al., 2017 is missing from the bibliography. L161–163: Why was the SNR computed for the industrial era for the mass balance but not for the length fluctuations? L169–171: There appears to be a negative trend in L' for South Cascade at tau=30 as well. More generally, however, it seems too strong to claim an "absence of 20th-century retreat in the Northern Hemisphere" based on the sample of three glaciers. Do you find this also when you look at the larger sample of 76 glaciers? L242: Are the ratios similar when you choose a larger tau?

References: – Bach, E., Radić, V., & Schoof, C. (2018). How sensitive are mountain glaciers to climate change? Insights from a block model. Journal of Glaciology, 64(244), 247–258. https://doi.org/10.1017/jog.2018.15 – Raper, S. C. B., & Braithwaite, R. J. (2009). Glacier volume response time and its links to climate and topography based on a conceptual model of glacier hypsometry. The Cryosphere, 3(2), 183–194. https://doi.org/10.5194/tc-3-183-2009

---

## Author Comment (AC1) · 9 Dec 2020

**Response to comments from reviewer #1**

As it stands I find that the analyses are very well done and a lot of new insight is provided. This manuscript provides new theory for interpreting past glacial length changes. It uses an approach that honors understandings of climate variability that have been around in the atmospheric science community for decades but have not yet made their way into the glaciological or glacial geology communities.

With a bit of work to improve legibility, the manuscript can be a long-lasting contribution. Further synthesis of the results is needed. Because the implications of this work are most pertinent for folks working on paleoglaciers and the interpretation of the moraine record, the manuscript should be adjusted to make it more accessible for these folks. If not adjusted the important insights provided here might be missed or overlooked. Further suggestions to aid this effort are provided in the minor comments below.

We thank the reviewer for their thorough assessment of the manuscript and helpful recommendations for its improvement. We have substantially revised the manuscript to address the reviewer's concerns.

In some parts of the manuscript so much information is presented that the reader is overwhelmed. This is the case even for a reviewer who is familiar with the methods the author's apply. The analyses vary rapidly in spatial scope (3 glaciers, 76 glaciers, hemispheric) and temporal scope (pre-1880, post-1880) or vary between different components of the mass balance (summer, winter, annual) or the response time (10, 30, 200 yrs). While it is a good 'problem' to have too many analyses I think the manuscript would be more effective if there were a more central thread to follow.

We have made substantial changes to the introduction, transitions and figure captions, and hope that we've now given readers that central thread more clearly.

The introduction as it reads does not lay out the analyses to come. Removing some of the extraneous analyses and points would also help. The authors could also take a bit more space to explain their analyses/ results if more extraneous points were removed. Better transitions between sections 3.1-3.4 are also needed.

Thank you. We agree. We've rewritten the introduction to provide a clearer roadmap of the paper, and worked on the transitions between sections to guide a reader more clearly. We've also rewritten the figure captions to be clearer about what results are presented and how they should be interpreted.

The figures can also be overwhelming as they provide a lot of information. Again this is positive but the authors should consider how these figures could be simplified while still allowing the main points to come across. Perhaps some of the multi-paneled figures could be moved into the Appendix or Supplemental and be replaced with simplified versions in the main text to improve legibility?

We've added a new figure introducing spectral analyses, and rewritten all the figure captions to be clearer about what information is being presented. We've cleaned up the worst offender (the spectral figure), and throughout we've rewritten the presentation of the results, which we hope helps.

Along these lines the authors might consider removing the analysis related to individual forcings (section 3.4) and glacier length and save it for another manuscript. Figure 6 was interesting but I felt it was rather hard to take much away from it. From my view there are more than enough new contributions from Sections 3.1-3.3.

We appreciate the comment and suggestion, but we prefer to keep this figure in - the fact that it is volcanic forcing that is coordinating the coherent response of glaciers worldwide is, we think, a very important part of our analysis and conclusions. We hope that the revised text is substantially clearer now, and will not lead to information overload for a reader.

The manuscript could also be streamlined by narrowing the temporal scope. If the authors just analyzed 1000 to 1880 then any issues with the GCM output poorly representing the more recent climate would be removed or even put in the supplemental material. Furthermore, removing the 1880 to present time period would also lessen some of the more complex items the authors discuss in the manuscript now, allowing for a more streamlined read.

We appreciate the suggestion and carefully considered it. Ultimately we want to retain the industrial-era analysis (which only features at the start and at the end). We think readers are likely to be interested in the modern retreats, and furthermore, it is valuable to put preindustrial fluctuations and climate forcing in the context of modern industrial-era changes. We've added specific text to the figure captions and manuscript to be clear about the intervals being analyzed.

*The text should address the paleo glacier literature more directly. The paper dances around ideas related to the moraine record. I think the authors should meld in a bit of the analysis of Anderson et al., 2014 into the discussion. Who in their table 2 also show SNR as related to response time. While the Anderson paper uses the one-stage version of the linear model the same general trends are apparent with either the 1-stage or 3-stage. Their analysis if nothing else supports the conclusions of this paper: larger SNR results from longer glacier response times. Authors should also consider referencing Young et al. (2011) who at least note the potential role of varying response time in controlling the timing of moraine formation. The manuscript lays out some very important conclusions for those interpreting the moraine record/ paleo glacier lengths, but these points are not highlighted as much as they could at the end of the manuscript. Or they are overwhelmed by other points made in the discussion/conclusion section.

Thanks very much for this comment - this is really valuable. We have added in a discussion about the relevance of the results to interpreting the moraine record, and called for the need for a careful assessment of glacier dynamics when comparing moraines from different regions. We cite Young (thank you) as an example of where that was done. It is definitely a fruitful area for future research: we would like to add a moraine model into our analyses and evaluate moraine statistics as they covary between regions, along the lines of Anderson et al.

**Minor Comments:**

Line 20-21. A more accessible description of internal climate variability would be helpful here for those that do not think about climate in this way. Section 1 Introduction would benefit the reader more or a bit more by introducing say glacier response times and signal-to-noise ratios of glacier length/mass balance here. As it stands I was a bit surprised by the breadth of the analyses as I read down.

Thanks. We've added a bit more about internal variability and pointed readers to a reference. We added information in the introduction about the signal-to-noise ratio

Line 84. Perhaps a bit more about the data used to create these temperature anomalies would be useful here. Really just so the reader doesn't need to go looking.
We tweaked the text

Line 179-180 See Anderson et al., 2014 Table 2 for similar results.
Thanks for the reminder! We've added a note in the introduction about other uses of signal-to-noise in glaciology

Line 185. The extension of the analysis to 76 additional glaciers is a surprise. The reader would benefit from a bit of an outline of the simulations at the end of the introduction. Section 3.1 "Dependence of SNR on timescale" would be more clear as "Response time and SNR" same for Section 3.2 maybe "Spatial scale and SNR"
Thanks for this. We have rewritten the transition, and tweaked the section titles. We now try to clearly state that the reason for going to the full network of glaciers is that it allows us to evaluate the coherence both within and between different glacierized regions.

Line 196. I find this paragraph to be accurate but overwhelming if you are not already an atmospheric/climate scientist or previously familiar with the author's approach to linking climate variability to glacier response. So more simple explanations and simplifying will help.

Line 197. perhaps take a bit more space here to explain what 'white' noise means for those who are unfamiliar.
Thanks for these two comments. We've written an introduction to spectral analysis and now included a schematic figure illustrating how the spectra of climate variability and length fluctuations are linked.

Line 202. This sentence could be rewritten to just state what the source of the differences in slope are. Right now my brain is a bit overwhelmed looking at all the data in figure 3.
Thank you. We've rewritten the paragraph to lead a reader much more directly through the results.

Line 255-257. This is a really important observation one that helps bridge the gap between Quaternary geologists and the authors' more atmospheric science-based approach.
Thank you!

Line 274-276. This seems like a bit of an understatement on the part of the authors. There are scant few glaciers that have advanced through the last century.
We agree that glacier retreat has been pervasive. It was a phrasing issue that we've fixed.

Section 4 (Summary and Discussion) would be improved with more synthesis in how these results relate to folks who interpret moraine chronologies and past glacier fluctuations.
We've added a paragraph in the discussion talking about how to extend the results to moraine histories, and the impotence of understanding the glacier dynamics when comparing between settings.

For example: "The preceding analysis has shown that forced changes in glacier length are driven primarily by globally coherent changes in summer temperature." from line 239-240 would be good to emphasize here.

We added exactly this sentence in the discussion (thanks) , and also note that volcanic aerosols are responsible in this GCM.

Line 299. It would help to put a range of values of this in numbers as right now I can think of mountain ranges that vary in area by orders of magnitude. The 'individual mountain range' phrase was used above as well.
We changed the phrase to 'glacierized region', here and throughout. This phrase isn't exactly precise either, but we hope that the matrix, which we now group into different regions, gives a reader an idea of what we mean.

Line 300-302. The discussion from Anderson et al, 2014 is relevant here for moraine ages across the western US for the LGM. It is a real-world example that ties into a similar analysis. Its inclusion would add depth to the discussion.
Thanks, yes. We've included that. It would be really interesting to do more work like this.

Line 316. should be a new paragraph.
We moved this to earlier in the discussion and expanded it.

**Figures:**

Fig. 2. Perhaps the effect of glacier response time would be better shown if the y-axes were the same for all panels.
Thanks for this suggestion. We have standardized the y-axes across all rows, which makes it easier to compare the behavior of the different glaciers. However, we decided to continue using different y-axes for the different timescales, since this more clearly illustrates the point that different timescales yield different SNRs. It's true that timescale also affects the absolute magnitude of length variability, but that is not important here.

Fig. 3 Labels on one of the panels outlining what is low frequency and what is high frequency. Otherwise the reader needs to do a bit of math in their head if they are not used to looking at such figures. Perhaps the spectral slopes portion of the panels could each be in a box or subplot within the panel so they are differentiated from all of the other lines in each panel? The authors should consider de-emphasizing the data (maybe with transparency?) and emphasizing spectral slopes, which I find to be a more clear expression of the main point.
We've tweaked the figure. We've removed a couple of lines, lightened the grey lines, and thickened the colored lines, to try and enhance the clarity. We also changed the text describing the figure to be clearer about what we hope a reader gets from it. Finally, we hope that the addition of a schematic figure before this one, helps a reader appreciate the information more easily.

Fig. 4 This figure is again packed with information. While it is quite interesting and there is a clear trend I come away overwhelmed with information.
We've rewritten the figure captions, and the description in the text to try to be clearer.

Fig. 5 caption: "positive values indicate that summer temperature accounts for a larger fraction of variance in glacier length than in annual mass balance." I think the caption is incorrect here. As written temperature accounts for a larger fraction of variance in annual mass balance than in glacier length. I found that the text written about Figure 5 in section 3.3 was easier to follow than the figure itself. It

might be more effective for the authors to just describe the take home here in the text with a few statistics?

Thank you very much for catching the error! We've corrected panel (c), and have rewritten the caption to more clearly describe the contents and implications.

Fig. 6 I think this figure could be improved by defining the lines with different line styles. Right now the take away from this figure is not clear as well. In panels a and b, the lines are so dense that the reader has to work very hard to differentiate them. Maybe there is an easier way to present these results with a couple panels that are more legible or in a table with statistics?

We have made several changes that make the figure clearer. First, we corrected an error in the hemispheric averaging, which makes the lines less squiggly, especially in the Northern Hemisphere. Second, we bolded the lines and adjusted the aspect ratio of the figure, making the lines easier to see and compare. Finally, we added a vertical line at the pre-industrial/modern transition, which helps illustrate the difference in primary forcing agents between the hemispheres and eras. We hope these changes, along with revisions to the main text, make the figure and Section 3.4 much clearer.

---

## Author Comment (AC2) · 9 Dec 2020

**Response to comments from reviewer #2**

L25–29, 40–41: As far as I can tell from this discussion, these studies do not directly address glacier length variability, rather temperature variability. Please clarify this.
Thanks for pointing this out. We have changed the text to read "... was mostly attributed to natural forcing of temperature".

L47–53: Since the results of the paper rely on the fact that the ensemble spread can be used as a proxy for internal climate variability, I would like to see some more discussion of this. Have other studies used the LME for this purpose? How realistic is the magnitude of variability in LME? It would be good to cite some other papers that have similarly used ensembles of climate models for the purpose of disentangling forced and unforced variability.
Large initial condition ensembles have been widely used in other contexts to separate internal from forced variability. So our method is pretty standard; it's just the application of it to glaciers that is new. However, we agree it's important to provide more context. We have therefore added a citation of Deser et al., 2012 (Nature Climate Change) that explains this approach in detail.

Eq. 6: Please make clearer exactly how you compute this. The numerator is clear enough, but L90 "total variance across all ensemble members" is not totally clear.
Thanks for pointing this out. We have added the following parenthetical to the end of the sentence: (i.e., as if all 13 time series were concatenated into a single time series)

L140–144: There are a few studies that have looked at the dependence of response time on size on a global scale; see Raper and Braithwaite (2009), Bach et al. (2018).
Thanks, we have added these references.

L142: The reference Barth et al., 2017 is missing from the bibliography.
Thanks for pointing this out. We have added the reference.

L161–163: Why was the SNR computed for the industrial era for the mass balance but not for the length fluctuations?
The main reason we did not compute SNR values for glacier length over the modern period is that it is too short: the glacier length time series have too few degrees of freedom to compute SNR with any statistical confidence. However, we now state more explicitly in our revised draft that our focus is on glacier variability over the pre-industrial period.

L169–171: There appears to be a negative trend in L' for South Cascade at tau=30 as well. More generally, however, it seems too strong to claim an "absence of 20th-century retreat in the Northern Hemisphere" based on the sample of three glaciers. Do you find this also when you look at the larger sample of 76 glaciers?
We agree that "absence" is too strong here, and have replaced it with "lack". We do not find this behavior everywhere, but these two glaciers are fairly representative of the Northern Hemisphere overall. Figure A1 helps illustrate our main point, which is that the retreat is weaker than has been observed due in part to weaker-than-observed warming in the simulations over the last century in much of the Northern Hemisphere.

L242: Are the ratios similar when you choose a larger tau?

Good question. We have revised the paragraph and added two sentences which now address the dependence of our results on tau:

"Averaged across all glaciers, temperature accounts for 67\% of the total variance in annual mass balance, and 83\% of the total variance in glacier length when $\tau=10$ years. Temperature's share of the variance continues to increase with increasing $\tau$, but more modestly (to 86\% when $\tau=30$ years and 89\% when $\tau=100$ years)."